# Electronic Cigarette Use and Metabolic Syndrome Development: A Critical Review

**DOI:** 10.3390/toxics8040105

**Published:** 2020-11-17

**Authors:** Ilona Górna, Marta Napierala, Ewa Florek

**Affiliations:** 1Department of Bromatology, Poznan University of Medical Sciences, 60-354 Poznan, Poland; igorna@ump.edu.pl; 2Laboratory of Environmental Research, Department of Toxicology, Poznan University of Medical Sciences, 60-631 Poznan, Poland; martan@ump.edu.pl

**Keywords:** electronic cigarettes, metabolic syndrome, insulin resistance, dyslipidemia, obesity

## Abstract

The metabolic syndrome is a combination of several metabolic disorders, such as cardiovascular disease, atherosclerosis, and type 2 diabetes. Lifestyle modifications, including quitting smoking, are recommended to reduce the risk of metabolic syndrome and its associated complications. Not much research has been conducted in the field of e-cigarettes and the risk of metabolic syndrome. Furthermore, taking into account the influence of e-cigarettes vaping on the individual components of metabolic syndrome, i.e, abdominal obesity, insulin resistance, dyslipidemia and elevated arterial blood pressure, the results are also ambiguous. This article is a review and summary of existing reports on the impact of e-cigarettes on the development of metabolic syndrome as well as its individual components. A critical review for English language articles published until 30 June 2020 was made, using a PubMed (including MEDLINE), Cochrane, CINAHL Plus, and Web of Science data. The current research indicated that e-cigarettes use does not affect the development of insulin resistance, but could influence the level of glucose and pre-diabetic state development. The lipid of profile an increase in the TG level was reported, while the influence on the level of concentration of total cholesterol, LDL fraction, and HDL fraction differed. In most cases, e-cigarettes use increased the risk of developing abdominal obesity or higher arterial blood pressure. Further research is required to provide more evidence on this topic.

## 1. Introduction

In the European Union, around 700 thousand people die prematurely each year from the consequences of smoking tobacco [1]. Therefore, it is an important task of health policy to prevent people from taking up smoking and to encourage smokers to quit. Some data indicate that e-cigarettes can be used as a nicotine replacement therapy for smokers, but there is still insufficient evidence to support this hypothesis [1,2,3,4,5].

The basic questions are whether e-cigarettes are safe and whether they can cause chronic diseases. Although research so far has not provided a clear answer to these questions, experts believe that the negative effects of e-cigarettes use can only become apparent after many years. Initially, e-cigarettes were not subject to legal regulations, which resulted in significant inconsistencies in their composition (concentration of nicotine, tobacco-specific nitrosamines (TSNAs), heavy metals, polycyclic aromatic hydrocarbon (PAHs), other toxic and carcinogenic substances specific for tobacco) [6,7,8]. The currently applicable directive is more restrictive as for their composition [9]. Nevertheless, many other ingredients and supplements (such as vitamin E) that appear in e-cigarettes and their influence on human health have not been sufficiently studied. This, in turn, does not allow to explicitly affirm the safety of e-cigarettes [10]. 

Metabolic syndrome (MetS) is becoming more and more common among the citizens of highly developed Western countries. The results of a meta-analysis conducted by Sun et al. [11] indicate a clear connection between tobacco smoking and the risk of MetS. Based on data gathered in 13 prospective cohort studies (56,691 participants; 8688 cases of MetS), active tobacco smokers are at a 26% higher risk of MetS than nonsmokers [11]. In turn, there is still not much known about the influence of e-cigarettes use on the risk of developing MetS.

The present paper constitutes an overview and summary of the research reports so far on the topic of the influence of e-cigarettes on the development of MetS as well as its individual components. Moreover, the authors compared the actual knowledge about the impact of e-cigarettes use and smoking of conventional cigarettes (tobacco) on MetS.

## 2. Methodology

A critical review for 264 English language articles published until 30 June 2020 was made, using PubMed (including MEDLINE), CINAHL Plus, Cochrane, and Web of Science data. Twenty-three of these scientific papers were directly related to studies on e-cigarettes and the MetS and its individual components. Among them, 1 described an in vitro and animal study, 6 were experimental studies using animal models, and 16 studies concern the impact of e-cigarettes on their users. The literature search was performed using MESH terms and other relevant keywords. The key search terms were “electronic cigarettes”, “tobacco smoking” and “metabolic syndrome”, “insulin resistance”, “diabetes”, “dyslipidemia”, “obesity”, “blood pressure” (Table 1). Technical reports, expert statements, recommendations, other non-original papers as well as preprints and conference papers were excluded from our review. All of the presented original papers studies were peer-reviewed.

## 3. E-Cigarettes and Metabolic Syndrome

Metabolic syndrome among adults is diagnosed when three of the five following factors are present: abnormal waist circumference (depending on population group), triglyceride concentration (≥150 mg/dl or implemented hypolipidemic treatment), HDL cholesterol concentration (<40 mg/dl men, <50 mg/dl women or implemented treatment), arterial pressure (≥130/85 or implemented hypotensive treatment), glycaemia (≥100 mg/dl or implemented hypoglycemic treatment). Furthermore, the International Diabetes Federation (IDF) proposed definitions of the MetS adapted to three age groups of minors: 6–10 years, 10–16 years, and over 16 years of age. According to the IDF guidelines, the diagnosis of MetS should not be made in children under 10 years of age. However, if the child is obese and the accumulation of adipose tissue occurs mainly around the waist, it is advisable to take intensive measures aimed at weight reduction. In children over 10 years of age, according to the IDF guidelines, MetS can be diagnosed when abdominal obesity is present (waist circumference above the 95th percentile for a given child’s age) and the presence of at least two factors: elevated triglyceride, decreased HDL-C levels—same for both sexes, increased blood pressure, increased fasting glucose. In clinical practice, adolescents over 16 years of age use the adult MetS criteria [12,13,14,15,16,17,18].

MetS is diagnosed in 1/4 of the adult population worldwide [13]. Simultaneously, an increase in the number of cases among younger people has been observed in the last couple of years, especially in developing countries [14]. Research results indicate that MetS occurs in obese (60%) and overweight people (22%), but also in people with correct body weight (5%) [15]. In order to minimize the risk and complications of MetS, it is advisable to modify one’s lifestyle, increase physical activity, and quit smoking. Moreover, it has been demonstrated that the odds ratio of occurrence of MetS decreases with time, once the person has quit smoking [19,20].

The influence of smoking conventional cigarettes on the occurrence of MetS has been the subject of research for many years. The first mentions that some features of MetS stem from the smoking addiction that appeared in 1998 [21]. Since then, the results of numerous studies have pointed to an increase in the risk of developing MetS in active and second-hand tobacco smokers [22,23,24,25,26,27,28]. As studies have revealed, nicotine has the ability to inhibit lipogenesis, increase lipolysis as well as the amount of free fatty acids, which leads to the development of insulin resistance and type 2 diabetes mellitus. [29,30,31,32,33,34,35]. The risk assessment of occurrence of serious metabolic effects in the case of e-cigarettes use has not been thoroughly analyzed yet. However, taking into account current in vitro and in vivo studies using animal models, we can claim that even those e-cigarettes, which do not contain nicotine, can contribute to the changes in body weight, lipid profile, hyperglycemia, as well as in the level of oxidative stress [29,30,33].

So far, there have been very few publications demonstrating the influence of e-cigarettes on the risk of occurrence of MetS in people (Table 2). Research on a group of 18,300 volunteers was conducted by Lequy et al. [36]. The obtained results allowed the researchers to claim that the occurrence of MetS was connected with using e-cigarettes, whenever patients were not on a diet. Another study [37] has been conducted on a group of 7505 Korean males over 19 years of age to compare the frequency of occurrence of cardiovascular risk factors among: males smoking conventional cigarettes, dual users (e-cigarettes and traditional cigarettes users) and never-smokers. The multivariate logistic regression analysis showed that, in the adjusted model, the odds ratio (OR) for the prevalence of MetS equaled 2.79 (*p*  <  0.001) in comparison with never-smokers and 1.57 (*p*  =  0.038) in relation to cigarette-only smokers [37]. In this model, dual users were characterized by an increased waist circumference and TG level, as well as reduced HDL in comparison with both the people who had never smoked and those who smoked only traditional cigarettes [37]; therefore, it may be concluded that dual users are more vulnerable to cardiovascular risk factors.

In the newest study presented by Oh et al. [38], the type of smoking (conventional ones versus dual users) on the risk of the occurrence of MetS was assessed as well as its relation with the sex of the examined persons. The analysis of smoking history involved 5462 cases of MetS and a control group of 12,194 people. The obtained results demonstrated a lack of influence of the smoking kind on the occurrence of MetS among men as well as among women. However, the increase in the risk of the occurrence of MetS was related to smoking a packet of cigarettes for 20 years or more [38].

## 4. Insulin Resistance

Insulin resistance (IR) is the main pathogenic factor in type 2 diabetes mellitus, prediabetic state, as well as lipodystrophy [39]. There are many causes of insulin resistance, the main one being excessive body weight. Research results suggest that values of waist circumference over 94 cm in men and 80 cm in women belong to the first symptoms of insulin resistance. Frequently, the accompanying symptoms include increased fasting blood glucose level, arterial hypertension, and hyperlipidemia [39,40].

Studies have shown that nicotine causes decreased sensitivity of tissues to insulin, thus increasing the risk of developing type 2 diabetes mellitus [41]. Nicotine contributes to an increase in the level of hormones antagonistic to insulin, i.e, catecholamines and cortisol [42]. In turn, research studies conducted on animals have shown a direct influence of nicotine on activation of protein kinase dependent on AMP in adipose tissue. It increased the speed of lipolysis and promoted insulin resistance [32]. Insulin resistance, which causes the development of diabetes mellitus, is connected with prothrombotic inflammatory conditions and provokes atherogenic changes in blood lipids. It increases the risk of ischemic heart disease, diseases of peripheral arteries, and brain stroke [43].

The research conducted by Orimoloye et al. [44] concerned electronic cigarettes and insulin resistance in animals and humans. Their results were based on a controlled animal study and the National Health and Nutrition Examination Survey (NHANES 2013–2016) that involved 3989 participants. The assessed criterion was the influence of tobacco smoking and e-cigarettes use on the occurrence of insulin resistance, modelled with the use of homeostasis model assessment-insulin resistance (HOMA-IR) and glucose tolerance test (GTT). Research conducted on C57BL6/J mice exposed to e-cigarette aerosol or mainstream cigarette smoke for 12 weeks did not demonstrate any significant differences in comparison to the control group (filtered air) [44]. An analysis of the NHANES data also demonstrated no correlation between decreased sensitivity of tissues to insulin and different types of smokers (conventional cigarettes smokers, e-cigarettes users, dual users and nonsmokers) [44]. A lack of significant differences in the level of glucose was demonstrated also by Kim et al. [37] in their aforementioned study. 

Research on the influence of nicotine on the organism has demonstrated its ability to induce hyperglycemia by activating glycogenesis and gluconeogenesis [45]. El Golli et al. [46] conducted experiment on rats administered intraperitoneally exposed to pure nicotine (0.5 mg/kg of body weight), e-liquid with or without nicotine during 28 days [46], and a significant increase in the level of blood glucose was reported. Additionally, a reduced level of liver proteins and increased transaminase activity were detected in all the examined groups, yet it was the rats exposed to e-cigarettes which achieved worse results than the rats exposed to pure nicotine. It can suggest that the mechanism of disorders of glucose metabolism induced by e-cigarettes is different than in the case of nicotine alone [46].

Also, the connection between e-cigarettes use and prediabetic states has to be borne in mind. Research within this field has been conducted by Atuegwu et al. [35] among a group of people who are currently e-cigarettes users, those who used them in the past, and never-users. It was demonstrated that the current e-cigarettes users were at a greater risk of being diagnosed with prediabetic state than those who had never used this kind of cigarettes. Importantly, no relationship was reported between being a former e-cigarette user and prediabetic state [35] (Table 3).

## 5. Dyslipidemia

In the first phase of development of many cardiovascular diseases, hyperglycemia leads to damage to endothelium cells [49]. It is known that elevated glucose and insulin levels as well as insulin resistance can contribute to the development of atherosclerosis also through dyslipidemia, hypertension and inflammatory condition [50]. Increasing oxidative stress has an inhibitory effect on the availability of nitrogen oxide and creates advanced final products of glycation, which are significantly related to the risk factors of coronary disease and can favor its development [50]. Smoking cigarettes, obesity and IR influence a reduction in nitrogen oxide bioavailability. In turn, its elevated concentration diminishes adipose mass, concentration of glucose, triglycerides, free fatty acids and leptin in blood serum, increases mass of skeletal muscles and brown adipose mass [51]. Lipid disorders connected with insulin resistance manifest themselves mainly with elevated concentration of triglycerides (TG) and low concentration of HDL fraction cholesterol. Such abnormalities in the lipid profile contribute to an increased cardiovascular risk [40].

Numerous studies on the influence of smoking traditional cigarettes have revealed that higher concentrations of total cholesterol, TG, and LDL are reported in smokers, while a decrease in HDL fraction occurs [23,52,53]. Researchers link this correlation to a decreased activity of lipoprotein lipase and cholesterol acetyltransferase in plasma caused by smoking, as well as to an increased amount of free fatty acids. However, no changes in the influence on liver lipase have been reported [34]. Research has also demonstrated that nicotine causes redistribution of lipids and ectopic deposition of adipose tissue, thanks to which too-high TG concentration in the liver and blood serum leads to fatty liver [32]. Moreover, research conducted on mice which were given nicotine injections twice a day combined with a high-fat diet demonstrated an increase in oxidative stress, apoptosis activation and fatty liver [54].

Very significant research from the point of view of the influence of e-cigarettes on the state of the liver were conducted on mice by Hasan et al. [55]. The apolipoprotein E-deficient (ApoE-/-) mice were used in the study, because they develop fatty liver when fed a Western-type diet (Western Diet: 0.21% cholesterol, 21% kcal from fat, 50% kcal from carbohydrates and 20% kcal from proteins) at 45% of calories coming from fat. Moreover, this innovative model had all the features characteristic of nonalcoholic fatty liver disease as well as of MetS [55]. The mice were exposed to e-cigarettes and saline solution. The research results demonstrated that e-cigarettes caused an increased accumulation of lipids of diverse size in hepatocytes, while the number of cell organelles was reduced. Additionally, a significant increase in TG in the liver was reported [55]. Further research confirmed that e-cigarettes use led to NAD+ deficits. It was established that it can suggest a mechanistic relationship between damage to liver DNA caused by this type of cigarettes and mitochondrial dysfunction [56]. Similar results indicating a negative influence of e-cigarettes were obtained by El Golli et al. [57] in research conducted on rats treated for 28 days with intraperitoneal injections of pure saline solution, e-liquid without nicotine, e-liquid with nicotine content (0.5 mg/kg of body weight), and pure nicotine (0.5 mg/kg of body weight), which were diluted in saline solution. The analysis of the obtained results demonstrated lower oxidative stress and fewer histopathological disorders in the case of pure nicotine injections than in the case of e-liquids with nicotine. Furthermore, the authors observed increased activities of the liver biomarkers (aspartate aminotransferase, alanine aminotransferase, alkaline phosphatase, and lactate dehydrogenase) and identified inflammatory cells infiltration and cell death in the group exposed to e-liquid without nicotine. Therefore, they suggested that even e-liquids without nicotine can cause damage to the liver, and the addition of nicotine can enhance their adverse impact [57].

Also in another study, El Golli et al. [46] demonstrated that rats exposed for 28 days to e-cigarettes (both those containing nicotine: 0.5 mg/kg of body weight and those without it) showed significantly lower content of total cholesterol (TC) as well as of its LDL fraction in comparison to the group exposed to nicotine alone. Exposure to pure nicotine, in turn, caused an increase in TG in comparison with the other examined groups. In the case of the HDL fraction cholesterol, no reduction in comparison with the control group was reported, as opposed to the group exposed to nicotine, in which a decrease in HDL concentration was reported. The authors also demonstrated a significant increase in the TG/HDL ratio in the case of exposure to nicotine. In turn, for the TC/HDL and LDL/HDL ratios, lower values were reported during exposure to e-cigarettes, irrespective of whether they contained nicotine or not [46]. The authors point to the fact that, on the one hand, the improvement in lipid profile suggests a protective effect of e-liquid from e-cigarettes, yet on the other, they indicate that e-cigarettes can modify cholesterol content and thus influence hepatic function [46]. Due to the above, also liver biomarkers were subjected to an analysis. The analysis demonstrated a lower level of liver proteins as well as a higher alkaline phosphatase transaminases activity in all the examined groups, yet it was the rats exposed to e-cigarettes that showed greater deviations than those exposed to nicotine. The authors suggest that the abnormalities in the activity of liver enzymes are connected to damages to hepatocytes caused by e-cigarettes and nicotine [46].

It is also worth emphasizing that oxidative stress is of great significance. In case of its increase, the susceptibility of LDL fraction cholesterol to oxidation increases. Research conducted by Moheimani et al. [58] demonstrated an increase in oxidative stress in a group of persons who used e-cigarettes as opposed to nonsmokers. In turn, research conducted by Lerner et al. [59] reported similar reactivity of reactive oxygen species in e-cigarettes and cigarette smoke. Yet another study suggests that the level of oxidative stress can depend on the kind of liquid used in e-cigarettes. Having examined 11 liquids, the researchers concluded that only 1 was characterized by a significant increase in oxidative stress in human epithelium cell cultures [60]. However, it is worth highlighting that there is a great probability that the ingredients of liquids used in e-cigarettes will result in an increase in the number of reactive oxygen species once heated [58].

In their research on the myocardial infarction risk factors, Alzahrani et al. [61] analyzed electronic and traditional cigarettes. The participants of the study were nonsmokers, exsmokers, persons smoking from time to time, and daily smokers. Daily e-cigarette use as well as daily conventional cigarette smoking was independently associated with increased odds of having had a myocardial infarction. The authors demonstrated that odds of a myocardial infarction increased with elevated cholesterol concentration, occurrence of hypertension, and type 2 diabetes mellitus, as well as with age and sex. Although this study looked at risk factors for myocardial infarction, the relationship between e-cigarette smoking and increased cholesterol level, hypertension, and type 2 diabetes may be important for the risk of developing MetS [61]. 

Also, research conducted by Badea et al. [62] confirms the fact that e-cigarettes influence the lipid profile. A comparison between nonsmokers, people smoking traditional cigarettes, and those using e-cigarettes demonstrated that the LDL fraction concentration was significantly higher in the group smoking e-cigarettes than in the control group, as opposed to those smoking traditional cigarettes, in whose case a negligible increase was reported. The concentration of total cholesterol increased in all the groups of smokers, and despite the fact that no significant differences were demonstrated between them and the non-smoking group, a higher median value was reported in the group of people vaping e-cigarettes. In both groups also a slight decrease in the HDL fraction cholesterol was reported in comparison with the nonsmoking group, yet this decrease was greater in the group of people smoking e-cigarettes. In turn, in the case of VLDL, the group of e-cigarettes users demonstrated a smaller increase in its concentration in blood serum [62] (Table 4).

## 6. Abdominal Obesity

Obesity is characterized by an increase in body weight caused by a rise in the amount of adipose tissue (in men above 25%, in women above 30% of body weight) [63]. Adipose tissue does not only act as reserve supplies for the organism, but is also a source of bioactive agents, such as adipokines. Apart from their influence on the level of lipids, they are also responsible for inflammatory conditions, oxidative stress, insulin resistance and atherosclerosis [32,64]. Waist circumference is inextricably linked with the visceral adipose tissue content as well as the mass of total adipose tissue in the organism. Such accumulation of adipose tissue causes increased concentration of free fatty acids and leads to hyperinsulinemia and insulin resistance. In addition, it contributes to an increased risk of the occurrence of dyslipidemia, arterial hypertension, type 2 diabetes mellitus, cardiovascular diseases, and MetS [32,34,65].

Results of studies concerning Body Mass Index (BMI) indicate that its values are lower in smokers than in nonsmokers [66]. On the other hand, the authors point to the fact that an increase in BMI in the group of smokers can be related to the frequency of smoking and alcohol consumption as well as the lack of physical activity [66,67,68]. Moreover, the research demonstrated that nicotine decreases appetite and energy requirements as well as increases the rest metabolic rate [69]. Despite the fact that many studies indicate that BMI is lower in smokers than in nonsmokers, the distribution of adipose tissue is of key significance. An analysis of waist circumference only or the waist–hip ratio (WHR) instead of BMI suggests that smoking cigarettes favors greater accumulation of visceral adipose tissue [65,70].

For many years, research has been conducted on the influence of smoking on the development of abdominal obesity and higher WHR values [24,71,72,73,74,75,76]. Differences in the occurrence of abdominal obesity are observed also between the sexes [70]. According to researchers, the significantly more frequent central accumulation of adipose tissue in women connected with smoking can result from redistribution of tissue from the buttock and thigh area to the abdominal part or from amyotrophy [70]. Women are characterized by a greater content of adipose tissue, and that is why it is possible that nicotine causes greater fat utilization than in men. The mechanisms can concern differences in sex steroid levels in the serum, cortisol level in the plasma, occurrence of insulin resistance as well as other factors, i.e, the level of physical activity, diet, alcohol consumption, and stress level [68,70].

Research conducted on rats demonstrated that exposure to nicotine contributes to significant adipocytes hypertrophy with increased expression of transcription factors of proadipogenic genes. Such factors include the peroxisome proliferator activated receptor γ (PPAR-γ). The result of the aforementioned situation is an increase in body weight and accumulation of adipose tissue [77]. The creation of mature adipocytes is also influenced by oxidative stress [64,78]. Research conducted on mice demonstrated that during exposure to heavy smoking, increased activity of glutathione peroxidase in adipose tissue was reported, which can suggest an increase in oxidative stress inside it [79]. An analysis of other compounds that appear in cigarettes (i.e, carbon oxide, PAHs) also demonstrated their influence on increased lipolysis and adipokine secretion disorders. Such changes increase the risk of occurrence of cardiovascular diseases as well as metabolic diseases and their risk factors [80,81,82].

So far not much has been learnt about the influence of e-cigarettes on adipose tissue and adipocyte functions. The first such research was conducted by Zagoriti et al. [33]. It compared the influence of traditional cigarettes, e-cigarettes, and heated tobacco products on the differentiation of preadipocytes into beige adipocytes [33]. The increased activity of this kind of adipocytes (so-called thermogenic ones) was suggested as a means of reducing obesity and metabolic disorders. It was demonstrated that the extract of traditional cigarettes, as opposed to the other two extracts, significantly disrupted the differentiation of preadipocytes into beige adipocytes, which can influence the metabolic function of adipose tissue [33]. In turn, the results of research on e-cigarettes conducted on mice demonstrated that pure nicotine did not fully answer for lower body weight [83]. Further studies involving exposure of mice to equivalent nicotine doses, through inhaling e-cigarette or traditional cigarette fumes, did not show any influence on the body weight reduction in those exposed to e-cigarettes. It can suggest that it is other compounds in cigarettes that can participate in reducing body weight while smoking traditional cigarettes [84]. On the other hand, opposite results were obtained by El Goli et al. [48] and Werley et al. [85] in their research on rats.

The current possibility to substitute traditional cigarettes with e-cigarettes is responsible for the existence of the so-called “dual users”, who in want of quitting smoking start using e-cigarettes. In a study conducted by Kim et al. [37] among dual users and among persons smoking traditional cigarettes, it was shown that average waist circumference as well as the frequency of its occurrence were significantly higher than among people smoking both kinds of cigarettes. It should be emphasized that the long-term effects of such dual use are not well known yet.

Problems in establishing the influence of vaping e-cigarettes on obesity may stem from the fact that some people who are currently vaping e-cigarettes are former traditional cigarette smokers. Those errors were evaded in a research by Polosa et al. [86], who subjected people vaping e-cigarettes who had never smoked before to an observation that lasted 3.5 years. The obtained results did not demonstrate any significant changes in body weight in relation to the output value as well as in relation to the nonsmoking group. No significant differences were reported among people who used e-liquids containing nicotine either [86]. However, the very small size of the research group and the very young age of the researched persons were limiting factors of this study. On the other hand, Lanza et al. [87] indicated that e-cigarettes are growing in popularity, especially among young people. Members of this group often adjust their behavior to their peers in order to maximize social acceptance or minimize depressive disorders caused by peer exclusion [87]. In a study conducted by Delk et al. [88], the existence of positive correlation between the occurrence of obesity in a group of boys who smoked both traditional cigarettes and e-cigarettes was reported. However, no such correlation was reported in the group of girls. Disturbing reports concerning smoking among the youth are also provided by the research conducted by Cho et al. [89] among American teenagers. It was demonstrated that girls who perceived themselves as overweight significantly more often smoked e-cigarettes or were dual users [90].

It needs to be emphasized the vaping is a phenomenon that is currently causing controversies. There is scientific research that indicates it helps reduce the risk of increase in body weight in people who want to quit smoking and/or supports body weight control. Probably, it can stem from additional factors as well, such as taste, physical sensations in the mouth, and behavioral changes [90,91]. Perception of smoking as a factor that helps control body weight was analyzed in a study conducted by Rhoades et al. [92]. It was demonstrated that the claim that e-cigarettes help control body weight was rare. The most uncertain group included adults who smoked both e-cigarettes and traditional ones, though if such an observation was reported, it was more often among people who had used e-cigarettes [92] (Table 5).

## 7. Arterial Hypertension

Another criterion of MetS diagnosis is higher arterial pressure. Arterial hypertension is one of the most common factors that are responsible for the risk of development of heart conditions. The renin-angiotensin system is responsible for the regulation of arterial pressure. Many studies indicate that nicotine activates the sympathetic nervous system, releases noradrenaline and adrenaline, which cause nervous stimulation that can last up to 24 hours. Moreover, it inhibits aldosterone synthesis in adrenal glands while causing decreased aldosterone excretion [93]. An overview of sources available on the topic of the influence of traditional cigarettes indicates an increase in values of arterial pressure in smokers [24,94]. In the case of smoking e-cigarettes, the results are ambiguous.

The research conducted by Farsalinos et al. [94] demonstrated that short-term e-cigarette using caused a fractional increase in diastolic blood pressure. However, taking into account the time immediately after smoking, both the systolic and diastolic blood pressure, as well as the heart rate, were significantly elevated. According to the authors, nicotine from e-cigarettes was absorbed to a smaller degree, and that is why they did not show unfavorable influence on the heart function [94].

Also, in the study conducted by Yan et al. [93], a lower nicotine content was reported in the blood of persons vaping e-cigarettes than in the case of people smoking traditional cigarettes (Marlboro^®^), which contributed to a smaller increase in arterial pressure and heart rate in this group.

Research by Vlachopoulos et al. [95] was conducted among people of 30 ± 8 years of age without cardiovascular risk. The subject of analysis was arterial pressure measured in 4 sessions: smoking traditional cigarettes and e-cigarettes for 5 minutes, 30 minutes of vaping e-cigarettes, and sham procedure that lasted 60 minutes [95]. The analysis of the obtained results did not show any differences in all the readings between the sessions, while the systolic and diastolic arterial pressure values as well as the heart rate increased both during smoking traditional cigarettes and e-cigarettes [95].

Very significant results were obtained in the research conducted by Polosa et al. [86]. During a 3.5 years’ observation of a group of young people (29.7 ± 6 years of age), daily e-cigarette users, who had never smoked before, no significant changes in the heart rate and arterial pressure values were demonstrated in comparison with the control group, who had never smoked any kind of cigarettes [86].

In turn, in a study conducted by Franzen et al. [96], persons vaping e-cigarettes and traditional cigarettes were subjected to a 2 h observation immediately after having smoked a cigarette. The analysis of the obtained results demonstrated that systolic blood pressure increased for up to 45 minutes after the person had stopped using an e-cigarette containing nicotine and for ca. 15 minutes in the case of normal cigarettes. No significant changes in arterial pressure values in the first hour of observation were reported for e-cigarettes. A similar correlation was demonstrated for heart rate readings [96].

Research in that field was also conducted by Antoniewicz et al. [97] to assess acute influence of inhaling e-cigarettes with nicotine and without it within the period of 0, 2, and 4 hours after exposure. A significant increase in heart rate was reported in the case of vaping e-cigarettes containing nicotine and an increase in arterial blood pressure in both analyzed cases [97].

In the study of Arastoo et al. [98] among 100 electronic and traditional smokers, including chronic smokers, the baseline heart rate variability as well as hemodynamics (blood pressure and heart rate) were analyzed. In this research, the authors hypothesized that the changes in these parameters are caused by nicotine, and not by the non-nicotine components of the e-cigarette aerosol. Based on the conducted research, it was shown that people who were chronic smokers of both types of cigarettes had a similar level of heart rate variability. When assessing the effect of vaping e-cigarettes, a sharp increase in blood pressure and heart rate was noticed only after the use of e-cigarettes with nicotine, which confirmed the hypothesis [98].

An interesting study was also conducted by George et al. [99] on a group of people of ≥18 of age, who had been smoking ≥15 cigarettes for ≥2 years. During the study, the examined persons changed to using e-cigarettes with or without nicotine. The analysis of the obtained results demonstrated that once both groups of e-cigarette smokers had been combined, a greater decrease in systolic blood pressure was reported than in the group of traditional smokers.

Abnormalities in heart rate during vaping e-cigarettes in comparison with nonsmokers were also demonstrated by Moheimani et al. [58]. Importantly, the authors indicate that it was not caused by the nicotine content in the e-cigarette liquid, because it was untraceable in the plasma of the examined persons. At the same time, they emphasize that it is nicotine metabolites that can produce adverse effects leading to a greater risk of cardiovascular diseases among people smoking this type of cigarettes [58].

E-cigarettes are very often chosen when a person wants to quit smoking traditional cigarettes. In a 12 month prospective randomized research conducted by Caponetto et al. [100], the researchers subjected 300 participants divided into 3 groups to an analysis. Study groups were given nicotine refill cartridges: (i) 7.2 mg for 12 weeks; (ii) 7.2 mg for 6 weeks and 5.4 mg for another 6 weeks. The control one (iii) was given cartridges without nicotine for 12 weeks. The analysis of the obtained results demonstrated a decrease in the frequency of smoking in each group. However, what is important is the fact that within the duration of the study, no significant changes in systolic and diastolic pressure or heart rate were demonstrated. Moreover, no differences between the analyzed groups were revealed [100]. In turn, in a continuation of that study, conducted by Farsalinos et al. [101], an additional classification of constant smoking phenotype was made, in which people were divided into those who quitted smoking in full, those who limited the number of smoked cigarettes and those in whose case quitting ended in failure. It was demonstrated that persons who limited or quitted smoking and changed to e-cigarettes were characterized by lower systolic arterial blood pressure. This effect was especially visible in persons with elevated values at the beginning of the examination [101].

It has to be taken into account that e-cigarettes, by contributing to an increased sympathetic heart activity, can be also responsible for increased oxidative stress, which will cause an increased cardiovascular risk. Moreover, the increased oxidative stress and inflammatory conditions will induce the development of atherosclerosis (Table 6). 

## 8. Conclusions

Scientific evidence regarding the human health effects of e-cigarettes is limited. It is difficult to compare the traditional cigarettes with e-cigarettes due to the different mechanism of action (electronic mechanism), the form of delivery (vaping), and differences in the content of toxic and carcinogenic substances between tobacco and e-liquid. In comparison to traditional cigarettes, in e-cigarettes mechanism there is no burning. Therefore, it is difficult to consider a comparison of exposure to the tobacco smoke and aerosol generated by e-cigarette in the same volume or same exposure. However, e-cigarettes are considered as nicotine replacement therapy for traditional smokers; therefore it is important to compare the health effects of both forms of exposure. The World Health Organization (WHO) underlines that e-cigarettes still pose a significant health risk. While e-cigarette aerosols may contain fewer toxicants than cigarette smoke, studies evaluating whether e-cigarettes are less harmful than cigarettes are inconclusive. There is still a lack of evidence that they are safe during repeated inhalation in long-term use [5,6,100].

Taking into account all the reviewed data, there are very few studies on the influence of e-cigarettes use on the development of MetS and its components. Among 23 studies, 1 was combined in vitro and in vivo study [33], 6 experimental studies used animal models (Wistar rats, Apoe-/- mice, BALB/cJ mice, C57BL6/J mice) [44,46,55,83,84,85], and 16 studies concern the impact of e-cigarettes on their users (mostly adult participants aged 18–96 years) [35,36,37,38,58,62,86,88,93,94,95,96,97,98,99,101]. 

Taking into account the components of MetS, i.e. abdominal obesity, insulin resistance, dyslipidemia, and elevated arterial blood pressure, the results concerning the influence of e-cigarettes vaping are ambiguous. It follows from the analyzed studies that using e-cigarettes did not affect the development of abdominal obesity, insulin resistance, and elevated arterial blood pressure, but could influence the level of glucose and prediabetic state development, and increased risk of obesity and hypertension. In the case of lipid profile an increase in the TG level was reported, while the influence on the level of concentration of total cholesterol, LDL fraction, and HDL fraction differed. 

Most of the available articles reported are limited in their design, methodology, and the used exposure time and lack of long-term follow-up. Animals studies were mostly based on one sex (males). Cohort studies consist mainly of dual smokers (tobacco smokers and e-cigarettes users at once), making it difficult to assess the health effects associated only with e-cigarettes. However, given that most e-cigarette users are dual users (e-cigarettes users and tobacco smokers at once), many of the current research studies were conducted based on their observation. Moreover, some of them included just male participants.

Therefore, a clear need still remains for the development of new studies regarding e-cigarettes and their impact on MetS risk. Better assessment of e-cigarettes type and use, and further longitudinal studies, are needed to clarify this relationship.

## Figures and Tables

**Table 1 toxics-08-00105-t001:** Search strategy.

Objectives	(i) To assess the contribution of existing literature of association of electronic cigarette use and metabolic syndrome development as well as its individual components (insulin resistance, diabetes, dyslipidemia, obesity, blood pressure). (ii) To compare the impact of e-cigarettes use and smoking conventional cigarettes on metabolic syndrome (iii) To identify relevant information and outline existing knowledge (iv) To identify any gap in the existing research
Research question	(i) Does electronic cigarette use have a significant impact on metabolic syndrome development? (ii) Does electronic cigarette use have a significant impact on the occurrence of metabolic syndrome components (insulin resistance, diabetes, dyslipidemia, obesity, blood pressure)?
Keywords	(i) “electronic cigarettes” or “tobacco smoking” and “metabolic syndrome”; (ii) “electronic cigarettes” or “tobacco smoking” and “insulin resistance”; (iii) “electronic cigarettes” or “tobacco smoking” and “diabetes”; (iv) “electronic cigarettes” or “tobacco smoking” and “dyslipidemia”; (v) “electronic cigarettes” or “tobacco smoking” and “obesity”; (vi) “electronic cigarettes” or “tobacco smoking” and “blood pressure”

**Table 2 toxics-08-00105-t002:** The impact of electronic cigarettes use and tobacco smoke exposure on MetS.

Type of Study	Participants/Rodents/Type of Cells	Country	Exposure Assessment	Results	Reference
**E-CIGARETTES**					
Human	Participants of the constancies cohort (*n* = 18,300, aged 18–96 years), current users of e-cigarette and current or ex-smokers of tobacco.	France	Questionnaire	**Increased risk** of MetS associated with e-cigarette use only when not on a diet, *p* < 0.05.	Lequy et al. [36]
Human	Male participants (*n* = 7505, aged 19 years or older), users of e-cigarette or tobacco or dual users.	Republic of Korea	Urinary cotinine	**No differences,** however in the adjusted model odds ratio for MetS was 2.79 (*p* < 0.001) compared with never smokers and 1.57 (*p* = 0.038) compared with cigarette-only smokers.	Kim et al. [37]
Human	Participants (*n* = 17,656, aged 20–70 years), data from the Korea National Health and Nutrition Examination Survey, tobacco or dual users.	Republic of Korea	Questionnaire	**Increased risk** of MetS among women and among men using packet of cigarettes for 20 years or more, OR 4.02, 95% CI 1.48–10.93.	Oh et al. [38]
**TOBACCO SMOKE**					
In vitro-cells	Mouse embryo 3T3-L1 pre-adipocytes (ATCC^®^CL-173™, ATCC, Manassas, VA, USA).	United States	Exposure to cigarette smoke, electronic cigarettes, and heated tobacco products.	**Increased** metabolic activity after 24 h (*p* < 0.05) and 48 h (*p* < 0.01) of treatment of tobacco smoke.	Zagoriti et al. [33]
Animal	Male Wistar rats (young adults).	Jordan	Exposure to waterpipe tobacco smoke.	**Increased risk** of MetS - abdominal circumference (*p* < 0.0001), body weight (*p* = 0.02), systolic blood pressure (*p* < 0.0001), fasting blood glucose (*p* < 0.0001).	Al-Sawalha et al. [22]
Human	Male participants (*n* = 5697, aged 26–75 years).	Japan	Questionnaire	**Increased risk** of MetS *p* < 0.05 compared with nonsmokers.	Matsushita et al. [20]
Human	Participants (*n* = 3051, aged 20–60 years).	United States	Questionnaire	**Increased risk** of MetS: each year increase in cigarette smoking, odds ratio (95% CI), 1.00 (0.99, 1.02), between current smoker or past, 1.51 (0.80, 2.87) vs. 1.64 (0.97, 2.79).	Yankey et al. [23]
Human	Participants (*n* = 2212, aged 30–49 years).	Venezuela	Questionnaire	**Increased risk** of MetS *p* < 0.001 compared with nonsmokers, in the multivariate analysis: smokers: OR, 1.54; 95% CI, 1.11–2.14, *p* = 0.010.	Bermudez et al. [24]
Human	Participants (*n* = 694, aged 18–44 years).	Taiwan	Questionnaire	**Increased risk** of MetS males higher MetS prevalence than females *p* < 0.001.	Lin et al. [26]
Human	Participants (*n* = 430, aged 22.52 ± 0.40 years).	Chile	Questionnaire	**Increased risk** of MetS, linear regression model: MetS score *p* < 0.02.	Cheng et al. [27]
Human	Participants (*n* = 1637, aged 70.5–7.9 years).	Colombia	Questionnaire	**Increased risk** of MetS, OR = 1.5; 95% CI = 1.0–2.4, *p* = 0.034.	Barranco-Ruiz et al. [28]

**Table 3 toxics-08-00105-t003:** The impact of electronic cigarettes use and tobacco smoke exposure on insulin resistance and diabetes.

Type of Study	Participants/Rodents/Type of Cells	Country	Exposure Assessment	Results	Reference
**E-CIGARETTES**					
Animal	C57BL6/J mice	United States	Exposure to e-cigarette aerosol or mainstream cigarette smoke comparable with filtered air-exposed controls.	**No differences** of insulin resistance Modelled using the homeostatic model assessment of insulin resistance (HOMA-IR) and glucose tolerance tests, Glucose tolerance tests *p* = 0.17.	Orimoloye et al. [44]
Animal	Male Wistar rats	Tunisia	E-liquid with or without nicotine and nicotine alone (0.5 mg/kg of body weight) were administered intraperitoneally during 28 days.	**Increased risk** of diabetes Plasma glucose, *p* < 0.05.	El Goli et al. [46]
Human	Participants (*n* = 154,404 aged >18 years).	United States	Questionnaire	**Increased risk** of diabetestested blood glucose in past 3 years, history of prediabetes, compared to never e-cigarette users 1.97 (95% CI, 1.25–3.10) for current e-cigarette users, and 1.07 (95% CI, 0.84–1.37) for former e-cigarette users.	Atuegwu et al. [35]
Human	Male participants (*n* = 7505 aged >19 years)	Republic of Korea	Urinary cotinine	**No differences** Fasting plasma glucose, *p* > 0.05.	Kim et al. [37]
**TOBACCO SMOKE**					
Animal	Male C57BL/6J mice (8 week old).	China	Low dosages of nicotine (0.8 mg/kg/d), high dosages of nicotine (4 mg/kg/d), or saline (vehicle) using Alzet osmotic pumps.	**Increased risk** of insulin resistance Serum insulin, *p* < 0.05.	Wu et al. [32]
Human	Participants (*n* = 5931 aged 45–84 years).	Multi- Ethnic Study	Self-reported tobacco status and reclassified by urinary cotinine.	**No differences** of insulin resistance Insulin resistance biomarkers *p* > 0.10 **No differences** of plasma glucose Fasting plasma glucose, *p* > 0.05.	Keith et al. [47]
Human	Participants (*n* = 12 aged 54 ±10 years, diabetic).	Sweden	Nicotine 0.3 µg/kg/min or NaCl was infused (2 h) during a euglycemic hyperinsulinemia clamp (4 h).	**Increased risk** of insulin resistance, plasma insulin *p* < 0.0001 **Increased risk** of diabetes Plasma glucose *p* = 0.0039.	Axelsson et al. [48]
Human	Review—25 prospective cohort studies.			**Increased risk** of diabetes 24 reported adjusted RRs greater than 1 (range for all studies, 0.82–3.74). The pooled adjusted RR was 1.44 (95% confidence interval (CI), 1.31–1.58).	Willi et al. [31]
Human	Participants (*n* = 3051, aged 20–60 years).	United States	Questionnaire	**Increased risk** of diabetes Fasting plasma glucose *p* < 0.001.	Yankey et al. [23]

**Table 4 toxics-08-00105-t004:** The impact of electronic cigarettes use and tobacco smoke exposure on dyslipidemia.

Type of Study	Participants/Rodents/Type of Cells	Country	Exposure Assessment	Results	Reference
**E-CIGARETTES**					
Animal	Male Wistar rats	Tunisia	E-liquid with or without nicotine and nicotine alone (0.5 mg/kg of body weight) administered intraperitoneally during 28 days.	**No differences** of plasma HDL *p* > 0.05; **Increase** of plasma TG, *p* < 0.05; **Decrease** of plasma LDL, *p* < 0.01; **Decrease** of plasma total cholesterol, *p* < 0.0001.	El Goli et al. [46]
Animal	Male adult C57BL/6J Apoe-/- mice	United States	Exposed to saline or e-cigarettes with 2.4% nicotine aerosol for 12 weeks.	**Increase** of plasma TG, *p* < 0.05;	Hasan et al. [55]
Human	Participants (*n* = 150, middle aged)	Romania	Self-reports of the participants.	**Decrease** of plasma HDL, *p* > 0.05; **Increase** of plasma LDL, *p* < 0.05; **Increase** of plasma total cholesterol, *p* > 0.05.	Badea et al. [62]
**TOBACCO SMOKE**					
Animal	The human CETP transgenic mice (C57BL/6)	China	Exposure to either room air or cigarette smoke at five cigarettes/d and 5 d/wk for 12 weeks	**Decrease** of plasma HDL, *p* < 0.05; **Increase** of plasma LDL, *p* < 0.01.	Zong et al. [53]
Animal	Male C57BL/6J mice (8 week old)	China	Low dosages of nicotine (0.8 mg kg^−1^ d^−1^), high dosages of nicotine (4 mg kg^−1^ d^−1^), or saline (vehicle) using Alzet osmotic pumps	**Increase** of hepatic TG and serum TG *p* < 0.05	Wu et al. [32]
Human	Review			**Decrease** of plasma HDL- is the most widely documented lipid abnormality related to smoking; **Increase** of plasma total cholesterol.	Athyros et al. [52]
Human	Participants (*n* = 3051, aged 20–60 years)	United States	Questionnaire	**Decrease** of plasma HDL, *p* < 0.01; **Increase** of plasma TG, *p* < 0.001; **Increase** of plasma LDL, *p* < 0.01.	Yankey et al. [23]

**Table 5 toxics-08-00105-t005:** The impact of electronic cigarettes use and tobacco smoke exposure on abdominal obesity.

Type of Study	Participants/Rodents/Type of Cells	Country	Exposure Assessment	Results	Reference
**E-CIGARETTES**					
In vitro-cells	Mouse embryo 3T3-L1 pre-adipocytes (ATCC^®^CL-173™, ATCC, Manassas, VA, USA).	United States	Exposure to cigarette smoke, electronic cigarettes and heated tobacco products.	**No differences** in body mass, E-cigarettes and heated tobacco products impact on the differentiation of preadipocytes into beige adipocytes *p* < 0.01.	Zagoriti et al. [33]
Animal	Male Wistar rats	Tunisia	E-liquid with or without nicotine and nicotine alone (0.5 mg/kg of body weight) were administered intraperitoneally during 28 days.	**Increase** risk of obesity, *p* > 0.05.	El Goli et al. [46]
Animal	Neonatal C57BL/6J mice	United States	Exposure to e-cigarettes for the first 10 days of life; e-cigarette cartridges contained either 1.8% nicotine in propylene glycol (PG) or PG vehicle alone; plasma and urine cotinine measurements.	**No differences** for pure nicotine of body weight, *p* > 0.05.	McGrath-Morrow et al. [83]
Animal	Male BALB/cJ mice (Charles River, Calco, Como), 183 month-old.	Italy	Exposure to the smoke of 21 cigarettes or e- cigarette vapor containing 16.8 mg of nicotine delivered by means of a mechanical ventilator for three 30-min sessions/day for seven weeks.	**No differences** in body weight, *p* > 0.05.	Ponzoni et al. [84]
Animal	Male and female Sprague–Dawley rats	United States	Exposure to low-, mid- and high-dose levels of aerosols composed of vehicle (glycerin and propylene glycol mixture); vehicle and 2.0% nicotine; or vehicle, 2.0% nicotine and flavor mixture; plasma nicotine and cotinine and carboxyhemoglobin levels.	**Increased risk** of obesity, *p* < 0.05 (MarkTen^®^ prototype e-cigarette).	Werley et al. [85]
Human	Male participants (*n* = 7505, aged >19 years).	Republic of Korea	Urinary cotinine	**Increase** of waist circumference, 2.26, 95% ci = 1.31–3.91, *p* = 0.003.	Kim et al. [37]
Human	Participants (*n* = 16, aged ≥18 years), prospective 3.5-year study.	Italy	Questionnaire	**No differences** in body weight, *p* = 0.95.	Polosa et al. [86]
Human	Participants (*n* = 2733) students in the 7th, 9th, and 11th grades.	United States	Questionnaire	**Increase** of weight in boys AOR = 3.45, 95% CI = 1.34, 8.33; *p* < 0.05.	Delk et al. [88]
**TOBACCO SMOKE**					
Animal	Female and male Sprague Dawley OFA rat.	Switzerland	Nicotine or saline was infused subcutaneously via Alzet osmotic mini-pumps; cotinine and other principal metabolites of nicotine in the serum.	**Increase** postnatal body weight after perinatal exposure to nicotine, *p* < 0.05.	Somm et al. [77]
Animal	Participants (*n* = 69,000 men and women from 42 populations participating in the first WHO MONICA survey, aged 35–64 years).	Finland	Questionnaire	**Decrease of BMI**, *p* < 0.05.	Molarius et al. [66]
Human	Male participants (*n* = 1122, aged 19–102 years).	United States	Questionnaire	**Increase of** body mass and BMI, *p* < 0.05	Shimokata et al. [72]
Human	Participants (*n* = 1948, aged 50–79 years).	United States	Questionnaire	**Increase of** BMI and WHR, *p* < 0.05	Barrett-Connor et al. [71]
Human	Participants (*n* = 1281) data from the Prospective Population Study of Women in Gothenburg.	Sweden	Questionnaire	**Increase of** BMI, *p* = 0.001.	Lissner et al. [73]
Human	Participants (*n* = 2341, aged 55–85 years).	Netherlands	Questionnaire	**Increase of** WHR *p* = 0.01 and BMI *p* > 0.05.	Visser et al. [74]
Human	Participants (*n* = 9047, aged 16–74 years).	Scotland	Questionnaire	**Increase of** WHR and waist circumference among women, *p* < 0.001; **Decrease of** hip circumference among men *p* = 0.001 and BMI *p* > 0.05.	Akbartabartoori et al. [70]
Human	Participants (*n* = 6123, aged 35–75 years, Caucasians)	Switzerland	Questionnaire	**Increase of** abdominal obesity among men: *p* = 0.03 and women *p* < 0.01.	Clair et al. [75]
Human	Participants (*n* = 487,527, aged 30–79 years)	China	Questionnaire	**Increase** of WHR, waist circumference, and waist circumference /height ratio (WHTR) *p* < 0.001.	Lv et al. [76]

**Table 6 toxics-08-00105-t006:** The impact of electronic cigarettes use and tobacco smoke exposure on arterial hypertension.

Type of Study	Participants/Rodents/Type of Cells	Country	Exposure Assessment	Results	Reference
**E-CIGARETTES**					
Human	Participants (*n* = 42, aged 21–45 years).	United States	Questionnaire, plasma cotinine	**Increase** of heart rate, mean SEM 1.37 [0.19], *p* = 0.05.	Moheimani et al. [58]
Human	Participants (*n* = 16, aged ≥18 years) prospective 3.5-year study.	Italy	Questionnaire	**No differences** of blood pressure and heart rate, *p* > 0.05.	Polosa et al. [86]
Human	Participants (*n* = 23, aged 23–58 years).	United States	Two commercial products that contain 16 mg/mL (1.6%) nicotine and three non-commercial products that contain 24 mg/mL (2.4%) nicotine, in the cartomizer device format attached to rechargeable batteries. In comparison Marlboro Gold King Size 0.8 mg per cigarette.	**Increase of** systolic (*p* = 0.04) and diastolic (*p* = 0.0001) blood pressure (five e-cigs vs. one Marlboro^®^ cigarette).	Yan et al. [93]
Human	Participants (*n* = 76, aged 36 ± 5 years).	Greece	Commercially-available tobacco cigarette: nicotine (1.0 mg), tar (10 mg) and carbon monoxide (10 mg) yields, and electronic cigarette, commercially-available device, with liquid containing 11 mg/mL nicotine concentration.	**Increase of** systolic (*p* < 0.001) and diastolic (*p* = 0.079) blood pressure; **Increase of** heart rate (*p* < 0.001).	Farsalinos et al. [94]
Human	Participants (*n* = 24, aged 30 ± 8 years) smokers free of cardiovascular risk factors	Greece	4 separate occasions (total 96 sessions): (1) tobacco cigarette over 5 min; (2) e-cigarette over 5 min; (3) e-cigarette for a period of 30 min and (4) nothing (sham procedure) for 60 min.	**Increase of** systolic and diastolic blood pressure *p* > 0.05; **Increase of** heart rate *p* < 0.05.	Vlachopoulos et al. [95],
Human	Participants (*n* = 15, aged 22.9 ± 3.5 year).	Germany	(1) Smoking a cigarette and inhaling into the lungs, (2) Vaping an e-cigarette with nicotine and (3) Vaping an e-cigarette without nicotine.	**Increase of** systolic and diastolic blood pressure *p* < 0.05; **Increase of** heart rate (with nicotine e-cig) *p* < 0.05.	Franzen et al. [96]
Human	Participants (*n* = 17, aged 26 ± 3 year).	Sweden	The e-liquid base consisted primarily of 49.4% propylene glycol, 44.4% vegetable glycerin, and 5% ethanol without any added flavorings. Premixed e-liquids with and without added nicotine were used (19 mg/mL and 0 mg/mL resp). Inhaled 30 puffs from the e-cigarette for 30 min, with each puff lasting approximately three seconds.	**Increase of** systolic and diastolic blood pressure, *p* < 0.001; **Increase of** heart rate, *p* = 0.015.	Antoniewicz et al. [97]
Human	Participants (*n* = 100, aged 21–45 year).	United States	E-cigarette-users after using an e-cigarette with nicotine, e-cigarette without nicotine, nicotine inhaler, or sham-vaping (control); nicotine and cotinine plasma levels.	**Increase** of systolic *p* = 0.0001 and diastolic *p* = 0.002 blood pressure, **Increase** of heart rate *p* < 0.0001.	Arastoo et al. [98]
Human	Participants who were willing to quit smoking (*n* = 114, aged ≥18 years).	United Kingdom	E-cigarette containing 16 mg nicotine or nicotine-free e-cigarette plus nicotine flavoring.	**Increase** of blood pressure: ≤20 pack-years *p* = 0.59 but >20 pack-years *p* = 0.04; **Increase** of heart rate: ≤20 pack-years *p* = 0.03; **Decrease** of heart rate: >20 pack-years *p* = 0.02.	George et al. [99]
Human	Participants who were willing to quit smoking (*n* = 300 aged 44.0 ± 12.5 years) smoking cessation clinic	Greece	E-cigarette kit with “Original” (2.4% nicotine—Group A), or “Categoria” (1.8% nicotine—Group B), or “Original” without nicotine (“sweet tobacco” aroma—Group C) cartridges.	**Decrease** of systolic blood pressure, *p* = 0.004.	Farsalinos et al. [101]
**TOBACCO SMOKE**					
Human	Participants (*n* = 23, aged 23–65 years).	United States	Two commercial products that contain 16 mg/mL (1.6%) nicotine and three non-commercial products that contain 24 mg/mL (2.4%) nicotine, in the cartomizer device format attached to rechargeable batteries. In comparison to Marlboro Gold King Size 0.8 mg per cigarette.	**Increase** of systolic (*p* = 0.02,) and diastolic (*p* < 0.05) blood pressure, heart rate (*p* = 0.001) among Marlboro^®^ cigarette users (five e-cigs vs. one Marlboro^®^ cigarette).	Yan et al. [93]
Human	Participants (*n* = 2 212, aged 30–49 years).	Venezuela	Questionnaire	Increase of blood pressure compared with nonsmokers, *p* < 0.001.	Bermudez et al. [24]

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
