# Peer review of "Electronic Cigarette Use and Metabolic Syndrome Development: A Critical Review"

_toxics, 2020, doi:10.3390/toxics8040105_

Round 1
Reviewer 1 Report
This paper reviews the impact of e-cigarette use (and active smoking) on Metabolic Syndrome. I read this paper with great interest, as e-cigarette use is increasing, particularly among children and young adults. Furthermore, previous research has reported that active smoking is associated with an increased risk for Metabolic Syndrome, as well as the individual components of Metabolic Syndrome (insulin resistance, dyslipidemia, obesity, and hypertension). There is a lot going for this appear- e-cigarette use is a very timely issue and this review paper would be of high interest. However, I have some serious concerns about this paper.
First, the review does not seem to be conducted with scientific rigor. The methods are shockingly vague- there is no description of the eligibility criteria, how many were deemed eligible, if studies of children were included, reason for exclusions, etc. Furthermore, there is no summary of the included studies- animal models, human studies, sample sizes, ages,definitions of metabolic syndrome used, how outcomes were assessed, effect estimates, etc. Table 1, which is the only table in this entire paper, is completely insufficient. I would expect a flowchart of the study selection, a table summarizing all of the included studies, and table(s) and/or forest plot(s) and/or other figure(s) summarizing the results. On those grounds alone, this paper is not publishable in its current states and will require major revisions.
Second, I am concerned with the lack of clarity in this paper. The novel and interesting aspect of this paper is it is the first review to review the impact of e-cigarette use on metabolic syndrome. However, the authors weave findings about cigarette smoking and even secondhand smoke (e.g. reference 25, Xie et. al) into the paper without ever explaining the rationale. This is particularly confusing, since the authors cite a review of active smoking and metabolic syndrome by Sun et al. in the Introduction. There may be a rationale for re-reviewing active smoking and metabolic syndrome (e.g. it appears that the authors are citing papers that are more recent than those included in the Sun et al. review), but the authors never state so.
Third, the organization of the paper is poor. First, some sections are lacking (see comment about Methodology). Second, adding more tables/figures would help to facilitate Sections 3 through 7. These sections are long and cumbersome, and could be improved by including tables for reference, rather than having to read out the details of the selected studies. Third, please find a better way to organize animal versus human studies, and the mechanisms. Perhaps add another subheading (e.g. 4.1 Epidemiologic studies 4.2 Animal models 4.3 Mechanisms) or restructure the paper to discuss human studies and then animal studies. Make the transitions more clear and indicate in your tables which results are based on human versus animal studies.
Finally, please refrain from using overly dramatic statements in the paper. Example: “epidemic of our time”.
Additional comments below:
Title: I am wondering about the appropriateness of the title “Effects of exposure to electronic cigarette and tobacco products on metabolic syndrome development”. First, the phrase “exposure to electronic cigarette” makes it sound like you are looking at secondhand exposure to e-cigarettes. This should be rephrased as “E-cigarette use”. Second, please indicate in the title that this is a review. Third, I am still not sure as to why you included “tobacco products” in this review- since you cite a review (reference #11 by Sun et al.) which has already done so.
Abstract:
Line 10: “Experts believe that the negative effects of e-cigarettes vaping may only become apparent after many years” This is a bold claim. Do you believe this claim is supported in your paper?
The abstract does not describe the paper. The “introduction” is too long and does not connect the ideas together. The abstract then ends by mentioning their review. No discussion of the results or discussion. Confusing and not a great start to the paper.
Introduction
Line 29: Remove the phrase “the addiction”. Sounds contrived.
Lines 29-31: You state “Some data indicate that e-cigarettes can be used as a nicotine replacement therapy for smokers, but there is still insufficient evidence to support this hypothesis.” Discuss how e-cigarettes can be an effective tool for quitting, but that they are dangerous due to the 1) potentially higher doses of nicotine; and 2) individuals who may not have otherwise smoked are vaping instead.
Line 36: Spell out all acronyms on the first use. Do this throughout the entire paper.
Lines 38-39: Citation?
Line 42: Generally better to cite specific papers. Also, why cite this review if you are also reviewing the impact of active smoking on metabolic syndrome as well?
Methodology:
This section is entirely too vague. I am honestly shocked that this is all the information provided about the systematic review. Please address all of the following questions: What was the eligibility criteria? Were studies in children allowed? Was there any attempt to differentiate the type of e-cigarette product (e.g. brand, flavoring, usage, etc.)? Was e-cigarette use assessed by self-report or biomarkers? How many potentially relevant articles were identified? How many were deemed eligible for the review? Of the studies that were excluded, what was the reason (e.g. too broad, did not use a specific definition of MetS, etc.)? Was there any attempt to audit the articles selected for the review? Finally, include a flow chart describing the study selection process.
Furthermore, you need to include more information about the studies. You will likely need an entire table describing the included studies. Information should include country, sample size, ages of the participants, how exposure was measured, whether the outcome was continuous or dichotomous, the effect size, and p-value. Table 1 is completely insufficient.
Lines 58-62: this is the definition in adults. The definition in children varies. Please make it explicitly clear whether studies in children were included in this review.
Lines 58: I believe MetS is the preferred acronym, since MS generally refers to multiple sclerosis. Also, why provide the acronym if you go back and forth between stating “metabolic syndrome” and “MS” throughout?
Lines 90: Interesting that this paper looked at dual users. How many of the other papers looked at dual users?
Lines 146-185: Jumping between human and animal studies. I would suggest organizing this better for clarity. Perhaps epidemiologic studies first, and then discuss how the toxicological results are confirmatory.
Lines 166-175: This section is quite interesting. I think the question of whether it is nicotine, along, or all the constituents of tobacco products that drives these associations is an important distinction to make. The El Golli study looked at nicotine, e-cigarettes with nicotine and e-cigarettes without nicotine in rats. Did all three exposures results in higher blood glucose? Also, would this not be considered “secondhand exposure” rather “active smoking”?
Line 183: You state that the association between e-cigarettes/cigarettes vary by sex, race/ethnicity, etc. Table 1 is referenced, but I see no additional information to support this claim in the table. Please reconcile this.
Lines 256-264: While this cited paper is interesting, it does not lend itself very well to the discussion about “dyslipidemia” (the subheading). I would remove or focus only on the dyslipidemia results.
Conclusions:
After reading a hard to follow paper, I cannot say for certain whether the conclusions are fully supported. I sincerely think adding in tables/figures and cutting down on the word count in the review would help to make your points more clearly. Then, perhaps, the conclusions can be articulated more clearly.
Author Response
Response to the comments made by the Reviewers
Manuscript ID: toxics-943122
Type of manuscript: Review
Title: Effects of exposure to electronic cigarette and tobacco products on metabolic syndrome development
We would like to thank the reviewers for their careful review of our manuscript and for providing us with some suggestions to improve its quality. We have carried out a major revision of the manuscript and we believe the paper has been significantly improved.
According to the Reviewers’ suggestion, the manuscript has been carefully checked and corrected. The changes in the manuscript have been highlighted by the colour font.
Below we sequentially address all of the points raised by the Reviewers.
Reviewer #1:
Firstly, we would like to express our deepest thanks to the Reviewer for devoting time to reviewing our manuscript, the corrections, and suggestions. We have carried out a major revision of the manuscript and we believe the paper has been significantly improved.
The Reviewer's comment: Title: I am wondering about the appropriateness of the title “Effects of exposure to electronic cigarette and tobacco products on metabolic syndrome development”. First, the phrase “exposure to an electronic cigarette” makes it sound like you are looking at secondhand exposure to e-cigarettes. This should be rephrased as “E-cigarette use”. Second, please indicate in the title that this is a review. Third, I am still not sure as to why you included “tobacco products” in this review- since you cite a review (reference #11 by Sun et al.) which has already done so.
The authors’ answer: According to the Reviewer's suggestion, the changes have been made and highlighted by the colour font in the manuscript.
The Reviewer's comment: Abstract: Line 10: “Experts believe that the negative effects of e-cigarettes vaping may only become apparent after many years” This is a bold claim. Do you believe this claim is supported in your paper? The abstract does not describe the paper. The “introduction” is too long and does not connect the ideas together. The abstract then ends by mentioning their review. No discussion of the results or discussion. Confusing and not a great start to the paper.
The authors’ answer: According to the Reviewer's suggestion, the changes have been made and highlighted by the colour font in the manuscript.
The Reviewer's comment: Introduction. Line 29: Remove the phrase “the addiction”. Sounds contrive
The authors’ answer: According to the Reviewer's suggestion, the changes have been made.
The Reviewer's comment: Introduction. Lines 29-31: You state “Some data indicate that e-cigarettes can be used as a nicotine replacement therapy for smokers, but there is still insufficient evidence to support this hypothesis.” Discuss how e-cigarettes can be an effective tool for quitting, but that they are dangerous due to the 1) potentially higher doses of nicotine, and 2) individuals who may not have otherwise smoked are vaping instead.
The authors’ answer: According to the Reviewer's suggestion, the changes have been made and highlighted by the colour font in the manuscript.
The Reviewer's comment: Introduction. Line 36: Spell out all acronyms on the first use. Do this throughout the entire paper.
The authors’ answer: According to the Reviewer's suggestion, the changes have been made and highlighted by the colour font in the manuscript.
The Reviewer's comment: Introduction. Line 42: Generally better to cite specific papers. Also, why cite this review if you are also reviewing the impact of active smoking on metabolic syndrome as well?
The authors’ answer: According to the Reviewer's suggestion, the changes have been made and highlighted by the colour font in the manuscript.
The Reviewer's comment: Methodology:
This section is entirely too vague. I am honestly shocked that this is all the information provided about the systematic review. Please address all of the following questions: What was the eligibility criteria? Were studies in children allowed? Was there any attempt to differentiate the type of e-cigarette product (e.g. brand, flavoring, usage, etc.)? Was e-cigarette use assessed by self-report or biomarkers? How many potentially relevant articles were identified? How many were deemed eligible for the review? Of the studies that were excluded, what was the reason (e.g. too broad, did not use a specific definition of MetS, etc.)? Was there any attempt to audit the articles selected for the review? Finally, include a flow chart describing the study selection process.
The authors’ answer: According to the Reviewer's suggestion, we have introduced appropriate additions to the text of the manuscript.
The Reviewer's comment: Furthermore, you need to include more information about the studies. You will likely need an entire table describing the included studies. Information should include country, sample size, ages of the participants, how exposure was measured, whether the outcome was continuous or dichotomous, the effect size, and p-value. Table 1 is completely insufficient.
The authors’ answer: According to the Reviewer's suggestion, the changes have been made and highlighted by the colour font in the manuscript. The data in Table 1, in individual parameters, are arranged chronologically in ascending order. More information about the research is in the text of the publication.
The Reviewer's comment: Lines 58-62: this is the definition in adults. The definition of children varies. Please make it explicitly clear whether studies in children were included in this review.
The authors’ answer: According to the Reviewer's suggestion, the changes have been made and highlighted by the colour font in the manuscript.
The Reviewer's comment: Lines 58: I believe MetS is the preferred acronym since MS generally refers to multiple sclerosis. Also, why provide the acronym if you go back and forth between stating “metabolic syndrome” and “MS” throughout?
The authors’ answer: According to the Reviewer's suggestion, the changes have been made and highlighted by the colour font in the manuscript. The used terminology was corrected in the whole manuscript.
The Reviewer's comment: Lines 90: Interesting that this paper looked at dual users. How many of the other papers looked at dual users?
The authors’ answer: Given that most e-cigarette users are dual users (e-cigarettes users and tobacco smokers at once), some of the current research was conducted based on their observation. The review of current research showed that there were two papers looked at dual users: Kim et al. cited as [37] and Oh et al. cited as [38] in this paper.
The Reviewer's comment: Lines 146-185: Jumping between human and animal studies. I would suggest organizing this better for clarity. Perhaps epidemiologic studies first, and then discuss how the toxicological results are confirmatory.
The authors’ answer: Taking into account the components of the metabolic syndrome, this cited system was chosen on purpose. Initially, the focus was on research into insulin resistance [46] - these are the only studies on the effects of e-cigarettes on MetS. Then, studies on the effects of e-cigarettes on glucose levels are discussed [37 - human studies and 48 - animal studies]. The last paragraph - human studies [35] describes the prediabetic state.
The Reviewer's comment: Lines 166-175: This section is quite interesting. I think the question of whether it is nicotine, along, or all the constituents of tobacco products that drive these associations is an important distinction to make. The El Golli study looked at nicotine, e-cigarettes with nicotine, and e-cigarettes without nicotine in rats. Did all three exposures results in higher blood glucose? Also, would this not be considered “secondhand exposure” rather “active smoking”?
The authors’ answer: Yes, in the study Golli et al. [48] on rats exposed to pure nicotine, e-cigarettes containing nicotine, and e-cigarettes without nicotine, a significant increase in the level of blood glucose was reported. We have emphasized this fact in the text of the manuscript.
The Reviewer's comment: Line 183: You state that the association between e-cigarettes/cigarettes vary by sex, race/ethnicity, etc. Table 1 is referenced, but I see no additional information to support this claim in the table. Please reconcile this.
The authors’ answer: According to the Reviewer's suggestion, the changes have been made and highlighted by the colour font in the manuscript. We changed the place of referenced Table 2.
The Reviewer's comment: Lines 256-264: While this cited paper is interesting, it does not lend itself very well to the discussion about “dyslipidemia” (the subheading). I would remove or focus only on the dyslipidemia results.
The authors’ answer: According to the Reviewer's suggestion, the changes have been made and highlighted by the colour font in the manuscript. Although this study looked at risk factors for myocardial infarction, the relationship between e-cigarette smoking and increased cholesterol level, hypertension, and type 2 diabetes may be important for the risk of developing MetS. We have given an explanation in the manuscript.
The Reviewer's comment: After reading a hard to follow a paper, I cannot say for certain whether the conclusions are fully supported. I sincerely think adding in tables/figures and cutting down on the word count in the review would help to make your points more clearly. Then, perhaps, the conclusions can be articulated more clearly.
The authors’ answer: According to the Reviewer's suggestion, the changes have been made and highlighted by the colour font in the manuscript.
Dear Editors and Reviewers, we appreciate all your insightful comments. Thank you very much once again for your patience and consideration.
On behalf of all authors.

Reviewer 2 Report
The article is interesting and provides a good update on electronic cigarettes and metabolic diseases.
However, the last 3 paragraphs (Line 443 to 455), seem to be lost in the text. It is not clear to the reviewer the aim of that part of the article. Please improve this part of the article or remove it.
Author Response
Response to the comments made by the Reviewers
Manuscript ID: toxics-943122
Type of manuscript: Review
Title: Effects of exposure to electronic cigarette and tobacco products on metabolic syndrome development
We would like to thank the reviewers for their careful review of our manuscript and for providing us with some suggestions to improve its quality. We have carried out a major revision of the manuscript and we believe the paper has been significantly improved.
According to the Reviewers’ suggestion, the manuscript has been carefully checked and corrected. The changes in the manuscript have been highlighted by the color font.
Below we sequentially address all of the points raised by the Reviewers.
Reviewer #2:
Firstly, we would like to express our deepest thanks to the Reviewer for devoting time to reviewing our manuscript, the corrections, and suggestions. We have carried out a major revision of the manuscript and we believe the paper has been significantly improved.
The Reviewer's comment: However, the last 3 paragraphs (Line 443 to 455), seem to be lost in the text. It is not clear to the Reviewer the aim of that part of the article. Please improve this part of the article or remove it.
The authors’ answer: According to the Reviewer's suggestion, the manuscript has been carefully checked and corrected. We removed this part of the manuscript.
Dear Editors and Reviewers, we appreciate all your insightful comments. Thank you very much once again for your patience and consideration.
On behalf of all authors.

Round 2
Reviewer 1 Report
Thank for taking the time to make changes to the paper. The paper is much improved. However, my concerns were not fully addressed and I remain concerned about the purpose, rigor and overall organization of this paper.
My major concerns are below:
Methods: My previous comment was partially addressed. Table 1 now includes the search terms, which is helpful. However, the authors still have not addressed many of my questions from the inital review. Please address the following critical questions: Were studies in children allowed? Was there any attempt to differentiate the type of e-cigarette product (e.g. brand, flavoring, usage, etc.)? Was e-cigarette use assessed by self-report or biomarkers? How many potentially relevant articles were identified? How many were deemed eligible for the review? Of the studies that were excluded, what was the reason (e.g. too broad, did not use a specific definition of MetS, etc.)? Was there any attempt to audit the articles selected for the review?
Please see the attached PRISMA checklist, which may provide insight about other aspects to present in the methodology section.
Table 2: This table is more informative now. However, I do think the information could be presented better. First, I suggest each outcome have its own table (to correspond to each section). Second, the table still lacks critical information! This is a review; therefore, the exposure classification should be very clear and the effect estimates need to be presented. It is already difficult enough to “compare” e-cigarettes to traditional cigarettes. However, this review does not even show effect estimates s Finally, I strongly advise that the authors create a separate row for each of the parameters ("animal or human", “country”, “ages”, “exposure assessment [continuous or categorical]”, “effect estimate”, etc.).
Results: In my opinion, the individual sections 3-7 are still hard to follow. I think the wordiness of these sections could be reduced if the authors added more information to the previous Table 2 (see comment above), as well as Forest Plots of the results.
Discussion: Where is the discussion about how e-cigarettes compare to traditional tobacco products? In the Introduction, the authors state that a secondary objective of this review was to “compare the actual knowledge about the impact of e-cigarettes use and smoking of conventional cigarettes (tobacco) on metabolic syndrome.” Yet, there is no real comparison about the magnitude/direction of the effects, mechanisms, dose-response, ages impacted, etc. This may be too ambitious for a single review. There appears to be two options: 1) remove the comparison aim (essentially take out the sentence from the Introduction and remove the "tobacco smoke" column from Table 1) and touch on this; or 2) weave in a comparison of the associations, mechanisms of cigarettes and e-cigarettes throughout the entire paper. My two cents: I think the authors should remove the "comparison" aim and focus on e-cigarettes.

Author Response
Response to the comments made by the Reviewer
Manuscript ID: toxics-943122
Type of manuscript: Review
Title: Electronic cigarette use and metabolic syndrome development: a critical review
We would like to thank the Reviewer for careful review of our manuscript and for providing us with some suggestions to improve its quality. We have carried out a major revision of the manuscript and we believe the paper has been significantly improved.
According to the Reviewer’s suggestion, the manuscript has been carefully checked and corrected. The changes in the manuscript have been highlighted by the colour font.
Below we sequentially address all of the points raised by the Reviewer.
Reviewer 1:
Firstly, we would like to express our deepest thanks to the Reviewer for devoting time to reviewing our manuscript, the corrections and suggestions. We have carried out a major revision of the manuscript and we believe the paper has been significantly improved.
The Reviewer's comment: Methods: My previous comment was partially addressed. Table 1 now includes the search terms, which is helpful. However, the authors still have not addressed many of my questions from the initial review. Please address the following critical questions: Were studies in children allowed? Was there any attempt to differentiate the type of e-cigarette product (e.g. brand, flavouring, usage, etc.)? Was e-cigarette use assessed by self-report or biomarkers? How many potentially relevant articles were identified? How many were deemed eligible for the review? Of the studies that were excluded, what was the reason (e.g. too broad, did not use a specific definition of MetS, etc.)? Was there any attempt to audit the articles selected for the review?
The Authors’ answer: According to the Reviewer's suggestion, the changes have been made and highlighted by the colour font in the manuscript.
The Reviewer's comment: This table is more informative now. However, I do think the information could be presented better. First, I suggest each outcome have its own table (to correspond to each section). Second, the table still lacks critical information! This is a review; therefore, the exposure classification should be very clear and the effect estimates need to be presented. It is already difficult enough to “compare” e-cigarettes to traditional cigarettes. However, this review does not even show effect estimates s Finally, I strongly advise that the authors create a separate row for each of the parameters ("animal or human", “country”, “ages”, “exposure assessment [continuous or categorical]”, “effect estimate”, etc.).
The Authors’ answer: According to the Reviewer's suggestion, the changes have been made and highlighted by the colour font in the manuscript.
The Reviewer's comment: Results: In my opinion, the individual sections 3-7 are still hard to follow. I think the wordiness of these sections could be reduced if the authors added more information to the previous Table 2 (see comment above), as well as Forest Plots of the results.
The Authors’ answer: According to the Reviewer's suggestion, the changes have been made in the manuscript.
The Reviewer's comment: Discussion: Where is the discussion about how e-cigarettes compare to traditional tobacco products? In the Introduction, the authors state that a secondary objective of this review was to “compare the actual knowledge about the impact of e-cigarettes use and smoking of conventional cigarettes (tobacco) on metabolic syndrome.” Yet, there is no real comparison of the magnitude/direction of the effects, mechanisms, dose-response, ages impacted, etc. This may be too ambitious for a single review. There appear to be two options: 1) remove the comparison aim (essentially take out the sentence from the Introduction and remove the "tobacco smoke" column from Table 1) and touch on this; or 2) weave in a comparison of the associations, mechanisms of cigarettes and e-cigarettes throughout the entire paper. My two cents: I think the authors should remove the "comparison" aim and focus on e-cigarettes.
The Authors’ answer: According to the Reviewer's suggestion, the changes have been made and highlighted by the colour font in the manuscript: “Scientific evidence regarding the human health effects of e-cigarettes is limited. It is difficult to compare the traditional cigarettes with traditional cigarettes due to the different mechanism of action (an electronic mechanism), the form of delivery (vaping) and differences in the content of toxic and carcinogenic substances between tobacco and e-liquid. In comparison to traditional cigarettes, in e-cigarettes mechanism, there is no burning. Therefore, it is difficult to consider a comparison of exposure to the tobacco smoke and aerosol generated by an e-cigarette in the same volume or same exposure. However, e-cigarettes are considered as nicotine replacement therapy for traditional smokers, therefore it is important to compare the health effects of both forms of exposure. The World Health Organization (WHO) underlines that e-cigarettes still pose a significant health risk. While e-cigarette aerosols may contain fewer toxicants than cigarette smoke, studies evaluating whether e-cigarettes are less harmful than cigarettes are inconclusive. There is still a lack of evidence that they are safe during repeated inhalation in long-term use [5,6,100].”
Dear Editors and Reviewer, we appreciate all your insightful comments. Thank you very much once again for your patience and consideration.
On behalf of all co-authors, Yours sincerely,
Ewa Florek
